# Optimization of the Territorial Spatial Patterns Based on MOP and PLUS Models: A Case Study from Hefei City, China

**DOI:** 10.3390/ijerph20031804

**Published:** 2023-01-18

**Authors:** Ran Yu, Hongsheng Cheng, Yun Ye, Qin Wang, Shuping Fan, Tan Li, Cheng Wang, Yue Su, Xingyu Zhang

**Affiliations:** 1School of Economics and Management, Anhui Agricultural University, Hefei 230036, China; 2Institute of Land and Resources, Anhui Agricultural University, Hefei 230036, China; 3Wuxi Forestry Station, Wuxi 214000, China

**Keywords:** territorial spatial pattern, spatial scale, spatial layout, MOP model, PLUS model

## Abstract

Optimization of the territorial spatial patterns can promote the functional balance and utilization efficiency of space, which is influenced by economic, social, ecological, and environmental factors. Consequently, the final implementation of spatial planning should address the issue of sustainable optimization of territorial spatial patterns, driven by multiple objectives. It has two components—the territorial spatial scale prediction and its layout simulation. Because a one-sided study of scale or layout is divisive, it is necessary to combine the two to form complete territorial spatial patterns. This paper took Hefei city as an example and optimized its territorial spatial scale using the multiple objective programming (MOP) model, with four objective functions. A computer simulation of the territorial spatial layout was created, using the patch-generating land use simulation (PLUS) model, with spatial driving factors, conversion rules, and the scale optimization result. To do this, statistical, empirical, land utilization, and spatially driven data were used. The function results showed that carbon accumulation and economic and ecological benefits would be ever-increasing, and carbon emissions would reach their peak in 2030. The year 2030 was a vital node for the two most important land use types in the spatial scale—construction land and farmland. It was projected that construction land would commence its transition from reduced to negative growth after that time, and farmland would start to rebound. The simulation results indicated that construction land in the main urban area would expand primarily to the west, with supplemental expansion to the east and north. In contrast, construction land in the counties would experience a nominal increase, and a future ecological corridor would develop along the route south of Chaohu County–Chaohu Waters–Lujiang County–south of Feixi County.

## 1. Introduction

Economic and social development has always affected territorial spatial patterns. It is a major strategic deployment of China to establish a scientific and efficient territorial spatial planning system. In 2019, Several Opinions on Establishing and Supervising the Implementation of the Territorial Spatial Planning System was issued by the Central Committee of the Communist Party of China and the State Council. It has stressed the need to integrate the various plannings such as the main functional area planning, urban and rural planning, and land use planning into a unified territorial spatial planning to realize the “multiple-planning integration”. According to research on various types of planning, such as regional planning [1], urban planning [2], land use planning [3], landscape planning [4], habitat management planning [5], etc., the spatial planning system exhibits a good development history in the international arena. However, these spatial plannings are relatively independent, so there are many works of literature to discuss their convergence or integration. For example, Lopes et al. thought that city planning encompasses disciplines related to socio-economic, land-use, transport, environment, and others, but these disciplines face communication difficulties and objectives divergence due to contradictory interests and isolated evolution [6]. Sangawongse et al. took Wat Ket, Chiang Mai, Thailand as an example to study its transition between planning modes, from centralized planning to collaborative urban land use planning [7]. Kaczmarek et al. developed a machine learning approach for the integration of spatial development plans based on natural language processing [8]. In China, territorial spatial planning represents a significant innovation in terms of spatial planning. Due to the relative independence and even contradictions among various spatial planning strategies, their articulation or integration has been explored widely. Thus, the focus of territorial spatial planning is on dealing with the problems of excessive planning types, overlapping and even conflicting planning contents. Consequently, the final implementation of territorial spatial planning should be embodied in the sustainable optimization of territorial spatial patterns driven by multiple objectives [9].

Studies concerning the territorial spatial pattern are mainly categorized into “production–living–ecological space” and “ecological–agricultural–urban zone”, but the focus is still on the scale [10] and layout [11] in the spatial expression. In other words, the scale structure and spatial layout of territorial utilization serve as the basis for further implementation of “production–living–ecological space” [12] and “ecological–agricultural–urban zone” [13]. The optimization of the territorial spatial scale is primarily oriented to economic, social, ecological, and environmental objectives. The scale structure of territorial utilization is deduced by using regression prediction, Markov prediction, neural network, system dynamics, and other methods. While the single-objective oriented settings are mainly low carbon [14], ecosystem service [10], land suitability [15], etc., the dual-objective oriented settings include land capability and suitability [16], economic benefits and ecosystem services [17], eco-environmental constraints [18], etc. Multi-objective settings with more than three objectives are rarely found. A variety of spatial models are used to conduct a computer simulation in time and space to optimize the territorial spatial layout by setting space conversion rules usually categorized as two types. One is parallelism, which uses many juxtaposition rules based on the current spatial configuration [19]. The other is the progressive rules that simulate in steps, and the former step is the necessary condition for the latter [20]. Currently, the prevailing spatial models include Agent [21], CLUE-S (the Conversation of Land Use and its Effects at Small regional extent) [22], CA (Cellular Automata) [11], and other improved models based on them. There are many articles that address scale optimization based on objective realization, which is relatively easy to operate when the spatial layout is not considered. In recent years, increasing numbers of papers on spatial simulation have appeared, but most set transformation rules based on the spatial status quo. In spatial planning, however, the scale and layout of any one moment are interrelated, so a one-sided study of scale or layout is divisive to some extent.

Planning provides vision and scientific guidance for regional future development, and the adjustment of spatial scale and layout should meet more comprehensive developmental objectives as far as possible. At the same time, good planning should be combined closely with current natural, economic, and social conditions and avoid extensive demolition and construction. This paper, therefore, carried out a multi-objective prediction for the territorial spatial scale and then combined the objective prediction results with spatial driving factors based on the GIS (Geographic Information Systems) platform, to optimize the territorial spatial layout. In the setting of multiple-objective functions, minimum emission and maximum accumulation of carbon were added to the conventional goals of maximum economic and ecological benefits, considering the “double carbon” target of peak and neutrality. In the setting of spatial driving factors, POI datasets that could characterize geographical entities were added to the conventional nature and location traffic conditions. The research results can therefore provide an integrated concept for the study of scale and layout in spatial planning and are a reference for planners to adjust regional scale and layout.

For the core organization of the article, Section 3.1. is the optimization of the territorial spatial scale, and Section 3.2. is the optimization of the territorial spatial layout. Section 3.1.1. is the optimization schemes of four functions. Section 3.1.2. gives the setting of constraint conditions in the optimization schemes. Section 3.1.3. is the scale optimization result obtained by using LINGO12 software based on MOP model. Section 3.2.1. is spatial driving factors, including three types of basic driving factors and nine POI datasets. Section 3.2.2. is the accuracy test of PLUS model. Section 3.2.3 is the simulation result of space layout obtained by using ArcGIS10.2 software based on PLUS model.

## 2. Materials and Methods

### 2.1. Study Area

As the capital of Anhui Province, Hefei is one of the four major cities of science and education in China and a sub-center city of the Yangtze River Delta. Hefei reached an urbanization rate of 82.28%, with GDP exceeding 1 trillion yuan in 2020, ranking among the top 20 cities in China in terms of the two indicators for the first time. According to the seventh national census, the resident population of urban areas in Hefei surpassed 5 million in 2020, advancing into the ranks of megacities in China. However, a previous study suggested that the construction land in Hefei grew from 113,000 to 207,100 ha from 1997 to 2012, with total carbon emissions increasing from 4,462,100 to 16,567,300 tons [23]. With its rapid economic and social development, the city faces the serious challenge of optimizing land use structure, maintaining ecosystem balance, and protecting resources and the environment. In particular, the situation raises a great challenge for achieving the “double carbon” goals (peak carbon emissions before 2030 and reach carbon neutrality before 2060) and territorial spatial planning. Hefei City not only has to keep up the strong impetus of rapid economic development but must also consider ecological benefits and strive to achieve carbon peak and carbon neutrality. As a city facing all these challenges, Hefei can be considered a strong candidate for requiring optimization of territorial spatial patterns, which makes it the subject of our study. The location map of Hefei is shown in Figure 1.

### 2.2. Data Source

This paper uses statistical, empirical, land utilization, and spatially driven data for analysis, derived from the government yearbook, and the research of eminent institutions, platforms, and scholars. Socio-economic and energy data, obtained mainly from Anhui Statistical Yearbook, Hefei Statistical Yearbook, and China Energy Statistical Yearbook from 2011 to 2019, form the statistical data. Empirical data comprising coefficients of carbon emission, carbon accumulation, ecological benefits, and economic benefits are based on the published research results. The land utilization data consists of the remote sensing images of Hefei City in 2010 and 2018, acquired from the Resource and Environment Science and Data Center of the Chinese Academy of Sciences (http://www.resdc.cn (accessed on 30 November 2022)). The data with a resolution of 30 m were classified into six categories following the Chinese Academy of Sciences classification system: farmland, forest land, grassland, water bodies, construction land, and unused land. The 2010 remote sensing image was derived from Landsat-TM/ETM, and the 2018 image from Landsat 8, with less than 10% cloud cover. The accuracy of human–computer interactive interpretation was more than 90%, which met research requirements. The spatially driven data mainly covered the elevation and slope of the land, distances from the highways, town roads, railroads, district and county headquarters, and other points of interest (POI). Elevation data were obtained from Geospatial Data Cloud (http://www.gscloud.cn (accessed on 30 November 2022)); the slope was calculated from elevation; POI data were obtained from Amap; and other spatially driven data were obtained from OpenStreetMap.

### 2.3. Methods

Both the multiple objective programming (MOP) and patch-generating land use simulation (PLUS) models were used in the study. First, the territorial spatial scale was optimized using the MOP model with four objective functions—minimum carbon emission, maximum carbon accumulation, maximum economic benefits, and maximum ecological benefits. In the next step, computer simulation was carried out on the territorial spatial layout with the PLUS model, setting space conversion rules and using the scale optimization result as quantitative objectives. The MOP and PLUS models were implemented using LINGO12 and ArcGIS10.2, respectively.

#### 2.3.1. MOP Model

The MOP model provides an optimal solution for multiple objectives within a given area. Generally, it comprises two or more objective functions, decision variables, and several constraining conditions, as shown in the following equation:(1)$$\{\begin{array}{l} min\;F\left(X_{ce}\right)=\sum_{i=1}^{6}X_{i}\times ce_{i}\\ max\;F\left(X_{ca}/X_{ee}/X_{eb}\right)=\sum_{i=1}^{6}X_{i}\times ca_{i}/ee_{i}/eb_{i}\\ \sum_{i,j=1}^{n}a_{ij}X_{i}=\left(\leq ,\geq \right)\;Z_{j},\;\;\;X_{i}\geq 0 \end{array}$$
where *F*(*X_ce_*), *F*(*X_ca_*), *F*(*X_ee_*), and *F*(*X_eb_*) denote the four objective functions representing carbon emission, carbon accumulation, and ecological and economic benefits, respectively. The decision variable *X_i_* collectively denotes the area of farmland, forest land, grassland, water bodies, construction land, and unused land. Entries *ce_i_*, *ca_i_*, *ee_i_*, and *eb_i_* represent the coefficients of carbon emission, carbon accumulation, ecological benefits, and economic benefits of each land use type. *a_ij_* is the constraint coefficient corresponding to the *j*th variable in the *i*th constraint condition, and *Z_j_* is the constraint value. In terms of the objective function and constraint condition setting, the Markov chain [24] and GM (1,1) [25] were adopted to predict some indicators.

Since there is more than one function in multiple objective programming, it is usually impossible to find a point where all the objective functions reach their maximum/minimum values. Therefore, in the specific solution, it is necessary to adopt a compromise scheme to make each objective function as large/small as possible. The fuzzy deviation method of LINGO12 software [26] can be used to fuzzy the objective functions, transform the multiple objective linear problem into a single objective, and obtain the optimal solution to the problem.

#### 2.3.2. PLUS Model

The PLUS model is a CA-based simulation model of spatial land utilization with a promising application in optimizing the territorial spatial layout [20]. It contains two modules. The first is a rule-mining framework based on a land expansion analysis strategy that explores each land type’s expansion and driving factors individually through a random forest algorithm. The second is a CA module based on multi-type random patch seeds, incorporating a decreasing threshold mechanism.

1.Random Forest Classification for Dual-State Decisions

(2)$$P_{i,k}^{d}\left(x\right)=\frac{\sum_{n=1}^{M}I\left(h_{n}\left(x\right)=d\right)}{M}$$
where $P_{i,k}^{d}$ denotes the growth probability of cell *i* to covert from the original land use type to the target type *k*; the value of *d* is either 0 or 1, and a value of 1 indicates that there were other land use types that changed to land use type *k*, while 0 represents other transitions; *x* is a vector that consists of multiple driving factors; *I*(∙) is the indicative function of the decision tree set; *h_n_*(*x*) is the prediction type of the *n*th decision tree for vector *x*; and *M* is the total count of decision trees.

2.CA Model Based on Multi-Type Random Patch Seeds

(3)$$OP_{i,k}^{d=1,t}=\{\begin{array}{ll} P_{i,k}^{d=1}\times \left(r\times \mu_{k}\right)\times D_{k}^{t} & if\;\Omega_{i,k}^{t}=0\;and\;r<P_{i,k}^{d=1}\\ P_{i,k}^{d=1}\times \Omega_{i,k}^{t}\times D_{k}^{t} & all\;others \end{array}$$(4)$$\Omega_{i,k}^{t}=\frac{con\left(c_{i}^{t-1}=k\right)}{n\times n-1}\times w_{k}$$(5)$$D_{k}^{t}=\{\begin{array}{cc} D_{k}^{t-1} & if\;\left|G_{k}^{t-1}\right|\leq \left|G_{k}^{t-2}\right|\\ D_{k}^{t-1}\times \frac{G_{k}^{t-2}}{G_{k}^{t-1}} & if\;0>G_{k}^{t-2}>G_{k}^{t-1}\\ D_{k}^{t-1}\times \frac{G_{k}^{t-1}}{G_{k}^{t-2}} & if\;G_{k}^{t-1}>G_{k}^{t-2}>0 \end{array}$$
where $OP_{i,k}^{d=1,t}$ denotes the growth probability of cell *i* to covert from the original land use type to the target type *k* at iteration time *t*; $\Omega_{i,k}^{t}$ denotes the neighborhood effects of grid cell *i* at time *t*; $D_{k}^{t}$ denotes the inertia coefficient for land use type *k* at time *t*; *r* is a random value ranging from 0 to 1; $\mu_{k}$ is the threshold to generate the new land use patches for land use type *k*; $con\left(c_{i}^{t-1}=k\right)$ represents the total number of cells occupied by land use type *k* at the *t* − 1th iteration within the *n* × *n* window; $w_{k}$ is the variable parameter ranging from 0 to 1 among the different land use types, and the closer it is to 1, the stronger the expansion ability of the land use type; and $G_{k}^{t-1}$ and $G_{k}^{t-2}$ are the differences between the current amount of, and future demand for, land use type *k* at the *t* − 1th and *t* − 2th iteration.

To control the generation of multiple land use patches, a threshold descending rule for the competition process was proposed to restrict both the organic and spontaneous growth of all land use types.
(6)$$\mathrm{If}\;\overset{N}{\underset{k=1}{\sum }}\left|G_{c}^{t-1}\right|-\overset{N}{\underset{k=1}{\sum }}\left|G_{c}^{t}\right|<Step\;Then,l=l+1$$
(7)$$\{\begin{array}{c} Change\;P_{i,c}^{d=1}>\tau \;\;and\;\;TM_{k,c}=1\\ No\;change\;P_{i,c}^{d=1}>\tau \;and\;TM_{k,c}=0 \end{array}\;\tau =\delta^{l}\times r1$$
where *Step* is the step size of the PLUS model to approximate the land use demand; *δ* is the decay factor of decreasing threshold *τ*, which ranges from 0 to 1 and is set according to the expert; *r*1 is a normally distributed stochastic value with a mean of 1, ranging from 0 to 2; *l* is the number of decay steps; and $TM_{k,c}$ is the transition matrix that defines whether land use type *k* is allowed to convert to type *c*, and a value of 1 indicates permission, while 0 indicates no permission.

## 3. Results

### 3.1. Optimization of the Territorial Spatial Scale

#### 3.1.1. Optimization Schemes

Following the timing of the current territorial spatial planning, optimization schemes for four objective functions were developed, choosing 2025, 2030, and 2035 as target years. Scenarios developed for carbon emission/accumulation as well as ecological/economic benefits are as follows. Their coefficients are shown in Table 1.

##### Minimum Carbon Emission

It is calculated by multiplying the area of each type of land by its carbon emission coefficient. The carbon emission/absorption ability of farmland, forest land, grassland, water bodies, and unused land, which have relatively stable carbon source/carbon sink capacities, is influenced mainly by changes in their areas. However, the carbon source capacity of the land for construction activities is significantly affected by human activities. As expected, the carbon emission coefficient changes with economic and social development. Thus, the minimum carbon emission level was decided based on results from Yu and Tian’s research [27] that were based on land use activities in Hefei City.

##### Maximum Carbon Accumulation

Based on Zhao et al. [28], the carbon accumulation coefficient of each type of land was obtained by calculating the total amount of accumulation and dividing it by the area of the land. The land’s relatively stable carbon accumulation coefficient without any construction activities was set as a fixed value, while the accumulation coefficient of construction land for each target year was obtained using the GM (1,1) model based on the calculated historical value.

##### Maximum Ecological Benefits

We used the method of Xie et al. [29] to calculate the ecological benefit coefficient of each land type for the target years. Based on the value of the world’s ecosystem services put forward by Costanza et al. [30], Xie et al. surveyed 700 Chinese professionals with ecological backgrounds. Based on their observations, they developed a set of evaluation methods for ecosystem services matching the conditions of China, which is what we followed.

##### Maximum Economic Benefits

The economic benefits of farmland, forest land, water bodies, and construction land were respectively represented by the output values of the planting industry, forestry, fisheries, and other secondary and tertiary industries. Given the difficulty to determine the economic benefits of grassland, the research results of Feng et al. [31] were used as a reference. Moreover, the minimum value of 0.001 ten thousand CNY/ha·year was taken for unused land that did not generate economic benefits. Then, the GM (1,1) model was used to obtain the economic benefit coefficients for the target years, based on the historical data of economic benefits per unit area in Hefei City from 2010 to 2018.

#### 3.1.2. Constraints

The perspectives of area, development of society, ecology, and economy were considered for the optimization schemes to set the various constraints and are itemized and discussed below.

In the absence of administrative divisions for the land, the sum of the area of all land types should be equal to the total area. In other words, the total area of Hefei City should be *S* = *X*_1_ + *X*_2_ + *X*_3_ + *X*_4_ + *X*_5_ + *X*_6_, where *X*_1_ to *X*_6_ represents various land use types, same as *X_i_* in Equation (1).

In the current period of rapid urbanization, the area of farmland will not exceed what is considered in the base period within a short time. Hence, the farmland area in 2018, considered the base period, was set as the upper limit. The base area in the target year predicted by Markov, based on historical data, was selected as the lower limit. Thus, the farmland area in the target year is predicted by Markov ≤ *X*_1_ ≤ the farmland area in 2018.

The forest land significantly contributes to reducing carbon emissions, increasing carbon accumulation and ecological benefits. Thus, its area in 2018, considered the base period, was set as the lower limit. There is an upper limit of a 10% increase of the current area because of its small increment, i.e., the forest land area in 2018 ≤ *X*_2_ ≤ current area × 110%.

Because of the low demand for grassland and its slow decline in the past years, the constraint of grassland area was set by referring to that of farmland area, i.e., the grassland area in the target year predicted by Markov ≤ *X*_3_ ≤ the grassland area in 2018.

The area covered by water shows stable fluctuations, within 0.5%. Thus, the constraint was formulated by increasing or decreasing the current area by 0.5%, i.e., the current area × 99.5% ≤ *X*_4_ ≤ the current area × 100.5%.

The construction land area for each target year was formulated based on previous research results by Yu and Huang [25]. Specifically, the construction land areas, *X*_5_, in 2025, 2030, and 2035 were designated as 227,297.70, 231,223.59, and 225,642.60 hectares, respectively.

Similar to grassland, the constraint was set as the unused land area in the target year predicted by Markov ≤ *X*_6_ ≤ the unused land area in 2018.

According to the population sizes and urbanization rates over the years, the total population, the population density of non-construction land, and the population density of construction land in each target year were predicted by the GM (1,1) model to develop the constraint on the population. This is defined as the population density of non-construction land × (*X*_1_ + *X*_2_ + *X*_3_ + *X*_4_ + *X*_6_) + population density of construction land × *X*_5_ ≤ total population.

For grain security, the grain demand per capita and the multiple-cropping index were set as 0.3925 tons/person [32] and 1.57 [33], respectively. Under the premise that no significant changes are expected in agricultural technology, the grain production capacity in each target year was predicted by the GM (1,1) model. The average values of 68.03% and 67.24% over the years were taken as the grain self-sufficiency rate and the ratio of the planting areas of food crops and cash crops, respectively. Thus, the constraint on grain demand was formulated as total population × 0.3925 × 68.03% ≤ *X*_1_ × grain production capacity × 67.24% × 1.57.

Considering the effectiveness of green equivalent in evaluating the quality of the regional ecological environment, green equivalent ≥ 1 was set as the constraint [34]. After the calculation of the ecological service values of farmland, forest land, and grassland, the green equivalents of farmland (*GE_fa_*), forest land (*GE_fo_*), and grassland (*GE_g_*) were calculated, respectively. Moreover, the target-year forest coverage rate (*FCR*) was predicted based on the target value and the increment in the National Economic and Social Development Planning of Hefei City. Thus, the constraint was set as (*GE_fa_* × *X*_1_+ *GE_fo_*× *X*_2_ + *GE_g_* × *X*_3_)/*S* × *FCR* ≥ 1.

For sustainable social and economic development, the ecological and economic benefits were set to follow a yearly growth trend.

#### 3.1.3. Results of Scale Optimization

For the optimization scheme of a single objective function, different schemes will result in different territorial spatial scales (Figure 2).

Synthesizing these single objectives to make each function as large/small as possible, as mentioned in Section 2.3.1., the territorial spatial scale optimization results for the Hefei City for 2025, 2030, and 2035 were obtained using the fuzzy deviation method in the LINGO12 software (Table 2).

Overall, an upward trend in the economic benefits was predicted, consistent with the current strategic positioning and rate of development of the city. Ecological benefits and carbon accumulation would show steady growth, and the peak values of carbon emissions were estimated to appear in 2030. A positive development trend was found in all the land use structures, carbon sources/sinks, and ecological benefits, indicating that relevant national policies were actively implemented. To be more specific, the forest land area in the ecological land would keep increasing; the grassland area would basically stabilize after a temporary and gentle decline. The area covered by water was estimated to fluctuate and show relatively smaller growth. The unused land would experience a low base but an insignificant reduction. Correspondingly, carbon accumulation and ecological benefits would exhibit an upward trend, reflecting the contribution of Hefei City to the construction of ecological civilization in the new era. During the year 2030, an important node in the model, the construction land area was predicted to first grow in decreasing increments and later show a negative growth rate. However, in the meantime, the area of farmland would begin to rebound, and carbon emissions would reach their peak.

On the one hand, there would be a balance between land under construction and farming by ensuring the healthy and rapid development of the economy and society. The indices of growth would reflect the improvements, both in the efficient use of construction land and the increase in farmland productivity. On the other hand, it would witness the maximum carbon emissions in 2030 but will actively move towards the higher carbon neutrality target by 2060.

### 3.2. Optimization of the Territorial Spatial Layout

#### 3.2.1. Spatial Driving Factors

Nature and location traffic conditions constitute the basic foundation of the scope, spatial distribution, and spatial expansion of a region. Based on previous studies, three types of basic driving factors have been generally identified: elevation, slope, and proximity factors [35,36,37,38]. In addition to the elevation and slope, distance to the highways, town roads, railroads, and district and county headquarters were used to characterize proximity factors (Figure 3).

A point of interest (POI) is a point location with name, address, and category attributes; all geographical entities that can be abstracted as points can be referred to in terms of POIs. POI datasets can thus provide useful socioeconomic information for significant locations, especially in relation to hospitals, schools, banks, shopping malls, and other facilities closely related to people’s lives. In addition, the introduction of dynamic POI data can effectively improve the accuracy of land use simulation [38]. Based on the development status and the changing trend of Hefei City, we select nine categories of POI data, including medical institutions, education institutions, financial institutions, commercial buildings, public service facilities, entertainment facilities, food and beverage outlets, transport sites, and living service facilities. Figure 4 reflects the density distribution of these POIs after normalization.

#### 3.2.2. Accuracy Testing

The Kappa coefficient or the figure of merit (FOM) coefficient is commonly used to test simulation accuracy. The value of the former lies between 0 and 1, and the closer the value to 1, the higher the simulation accuracy. While a Kappa coefficient larger than 0.7 is acceptable, values larger than 0.8 are considered satisfactory [19]. The value of the FOM coefficient also ranges between 0 and 1, and a larger value suggests higher simulation accuracy. In general, the FOM coefficient of about 0.15 means that the test is reliable [39]. For improved validity and reliability of the simulation, we used both coefficients to test the accuracy of the results. According to the territorial spatial pattern of Hefei City in 2010, the simulation was conducted under the settings described above for 2018 and then compared with the situations in the same year. The Kappa coefficient of 0.9235 and the FOM coefficient of 0.2487 obtained in our calculations suggest high and acceptable simulation accuracy. Based on the territorial spatial pattern for the year 2018, a simulation was conducted to generate the territorial spatial layout of Hefei City for three years—2025, 2030, and 2035, and the results are discussed next.

#### 3.2.3. Results of Simulation Optimization

Figure 5 is the visualization result of spatial layout optimization based on all previous studies. It meets the function equation, transformation rules, and other research goals set above: i.e., these equations and rules can be used for setting during the planning process, to ensure that implementation follows the optimal direction.

Our results show that during 2020–2025 and 2025–2030, the main urban area of Hefei City will continue to display a high demand for construction land. The models also predict that the city’s expansion would mainly be directed westward, with some additional activities to the east and north. This pattern matches the proposed long-term development plan of the city. It is consistent with the proposed expansion of the Xinqiao Science and Technology Innovation Demonstration Zone of Shushan District, which envisages the concentration of large scientific installations. It incorporates the New Canal City and the Feixi Industrial City Integration Demonstration Zone as the pivots. Growth in the northern wing would be around the Xiatang Industrial New City and Beicheng New District of Changfeng County. To the east, the city would expand, with the Feidong Industrial New District as its pivot.

The spatial coverage of construction land in Lujiang County, Chaohu County, and the central and northern Changfeng County would undergo a small incremental growth, mainly concentrated around the county headquarters. The incremental growth in the forest land areas was predicted to be mainly focused on Lujiang and Chaohu counties, with the future functional activities dominated by ecological space. Furthermore, the forest land areas of western Shushan District and eastern Feixi County would see a nominal rise because of the existence of Dashu Mountain National Forest Park and Zipeng Mountain National Forest Park. As mentioned previously, 2030 would be a prominent node due to various significant changes. During this period, carbon emissions are expected to peak. The land area growth under construction activities would change from a reduced state to negative values, while the farmland area is expected to show a nominal increase. Eventually, with all these changes, the territorial spatial layout of the city would enter a stage of steady consolidation. During 2030–2035, the land area under construction in the main urban area would basically remain stable, with a small reduction in the southern Lujiang and Chaohu counties and the nearby Zipeng Mountain in Feixi County, which is dominated by ecological space. Based on the above simulation results, the areas extending from the southern Chaohu County to the south and west, connecting with the Chaohu Waters and Lujiang County and southern Feixi County, would form an ecological corridor of Hefei City as a whole.

## 4. Discussion

Although the scope of government power at varying organizational levels of unitary, federal countries may differ in detail, they generally have similar planning logic and execution frameworks [1,2,3,4,5]. There are many types of spatial planning, however, and this can lead to contradictions, and even conflicts, in the implementation process [6,7,8], as some departments may not consider the interests of others in their plans. The innovation of China’s territorial spatial planning lies in the construction of a multiple-planning integration system, which would integrate economic and social development goals and balance the interests of different departments. However, the public’s greatest concern is how the plan will be implemented, and whether it will promote economic and social development. A specific spatial pattern supported by spatial scale [10] and layout [11] must therefore be in place, especially at the implementation stage. There are many research results on spatial scale optimization, but one or two factors, among economy, ecology, environment, and low carbon, are often chosen in the selection of optimization objectives. Although it reflects pertinence, it is difficult to reflect comprehensiveness. Spatial layout optimization is usually dominated by the setting of spatial factors, which is mostly a direct simulation based on the spatial status quo. It is difficult to fully reflect the statistical objectives of economic and social development through the spatial layout simulation. There are, however, few in-depth studies that combine the two, despite the fact that a one-sided study of scale or layout is divisive, because the two coexist and echo each other; it is, therefore, necessary to combine them to form complete territorial spatial patterns.

This paper put forward an overall optimization idea covering spatial scale and spatial layout. Specifically, compared to the common single-objective [10,14,15] or dual-objective [16,17,18] oriented settings, more comprehensive objective functions were scientifically set to optimize the scale of territorial use in a broad and effective manner. In the next step, the parallelism rule [19] and progressive rule [20] were integrated to provide a reference for the simulation of the territorial spatial layout based on multiple objective constraints, basic driving factors, and POI driving factors. In other words, various modules to be considered during territorial spatial planning and implementation were explored to achieve the various functions. These functions included minimum carbon emission, maximum carbon accumulation, maximum economic benefit, and maximum ecological benefit. Whether the objective functions are realized is a matter for post evaluation: i.e., the effects of the goals achieved when the planning reaches a certain stage of implementation. This implementation process will be directly reflected in the evolution of the spatial layout, but the evolution of this layout must be based on the current situation and is limited by factors that include nature, location of transportation, and built facilities. In addition to clarifying the conversion rules, spatial driving factors should therefore be included. Elevation and slope are natural geographical factors, while proximity factors are mainly location traffic factors, which measure the distance between a block and main roads and nodes. POI data can be obtained in real-time through Amap, and all geographical entities that can be abstracted as points can be referred to in terms of POIs. This paper selects facilities that are closely related to people’s lives, because planning is a reflection of human behavior, and both development and conservation should be centered on people. Perhaps, for this reason, the development of international spatial planning has continued to focus on human settlements.

As territorial spatial planning focuses on an all-encompassing structure that incorporates land use, urban and rural planning, as well as other developmental activities, it requires multiple levels of planning and integration. Based on the results of available research and with the prospects of increased data availability in the future, the introduction of more constraints or conversion rules is worth trying to improve the results of the optimization. For example, a criterion for the evaluation of resource and environmental carrying capacity [40] could be introduced in spatial scale optimization. Suitability evaluation of territory space [41] in the simulation of spatial layout could be another useful step that can be introduced. Spatial driving factor selection, the introduction of space management and control boundaries of permanent basic farmland [38], and the ecological protection red line [42], following the space conversion rules, are the other inputs that will contribute to more scientific and efficient planning and a clearer implementation path. In addition, once planning has been implemented to a certain stage, it can be evaluated according to the method used in this paper. Through the implementation evaluation, the phased results of planning implementation can be compared with prior optimization results, and the difference can be evaluated, so as to realize the mutual transformation of theory and practice.

## 5. Conclusions

In this paper, four objective functions were developed to realize multi-objective optimization of territorial spatial scale in Hefei City using the MOP model. On this basis, space conversion rules were developed to perform computer simulations of the territorial spatial layout via the PLUS model. The main conclusions are summarized below.

According to the results of scale optimization obtained by the MOP model, the area of forest land, a significant ecological land, would continue to grow with a large increment. However, the area of unused land and those covered by grassland and water bodies would reach a stable state with some fluctuations with small changes. Regarding the two critical land types, i.e., the farmland and construction land expanding to farmland, the year 2030 would be an important node. After 2030, the area of construction land would change from decreased to negative growth, while the net area of farmland would start to increase. These predictions demonstrate that while guaranteeing sustainable economic and social development, Hefei City will also meet other favorable changes in ecological parameters as well as improve the carbon footprint. With the creation of more carbon sinks and with higher carbon accumulation, the ecological benefits would increase steadily. The simulations show that carbon emissions will peak in 2030.

As revealed by the simulation results obtained by the PLUS model, from 2020 to 2030, construction land in the main urban area of Hefei City would mainly expand westward and then towards the east and the north. However, at the county level, the area of construction land would undergo only nominal growth. After 2030, the territorial spatial layout of the city would step into a period of steady consolidation. Lujiang and Chaohu counties would develop as important ecological spaces, and future years will see the growth of an ecological corridor to the south of Chaohu County–Chaohu Waters–Lujiang County–south of Feixi County.

## Figures and Tables

**Figure 1 ijerph-20-01804-f001:**
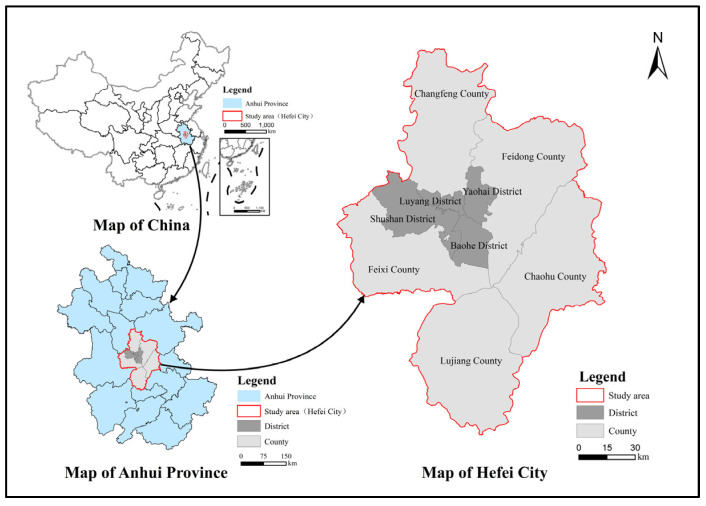
Geographical location of Hefei City.

**Figure 2 ijerph-20-01804-f002:**
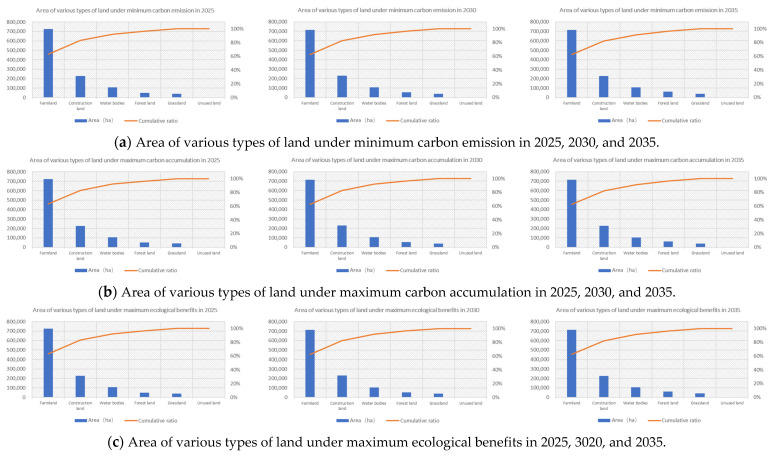
Area of various types of land under single objective optimization in 2025, 2030, and 2035.

**Figure 3 ijerph-20-01804-f003:**
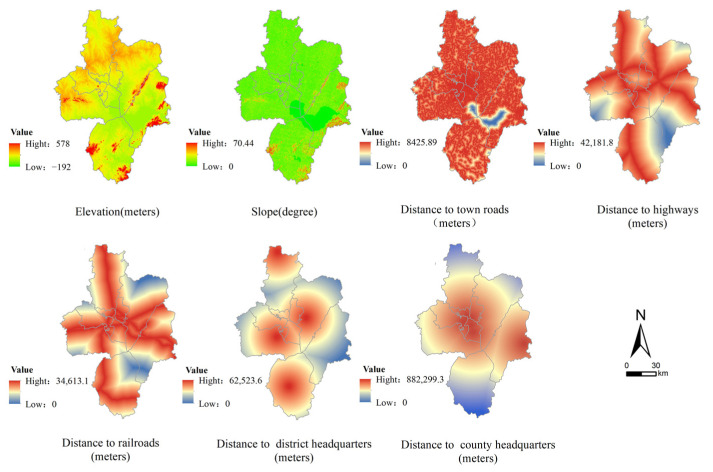
Basic driving factors of nature and location traffic conditions.

**Figure 4 ijerph-20-01804-f004:**
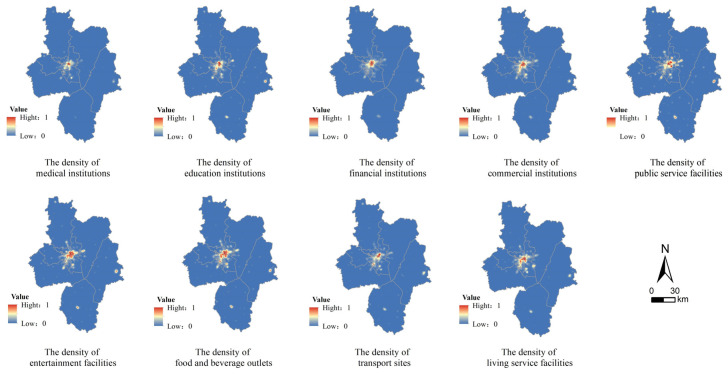
The density of selected POI driving factors.

**Figure 5 ijerph-20-01804-f005:**
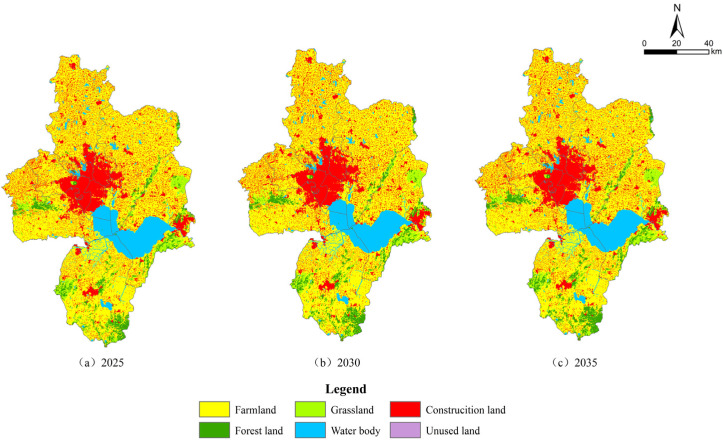
Simulation results of territorial spatial layout.

**Table 1 ijerph-20-01804-t001:** Coefficients of carbon emission/accumulation and ecological/economic benefits of various land use types in target years.

	Target Year	Farmland	Forest Land	Grassland	Water Body	Construction Land	Unused Land
carbon emission coefficient (t/ha)	2025	0.422	−0.644	−0.022	−0.253	169.900	−0.005
	2030					333.191	
	2035					332.136	
carbon accumulation coefficient (t/ha)	2025	63.015	114.331	43.965	22.690	89.436	66.973
	2030					98.693	
	2035					103.198	
ecological benefit coefficient (ten thousand CNY/ha·year)	2025	0.535	3.178	2.71	17.288	0.028	0.151
	2030						
	2035						
economic benefit coefficient (ten thousand CNY/ha·year)	2025	4.381	0.982	1.990	6.296	377.042	0.001
	2030	7.184	2.553		11.653	1362.647	
	2035	8.695	3.823		14.869	2379.650	

**Table 2 ijerph-20-01804-t002:** Optimization results of territorial spatial scales and objective functions under multiple objective programming.

	2018	2025	2030	2035
farmland (ha)	748,597.50	723,702.87	714,912.12	717,800.85
forest land (ha)	48,031.65	49,539.15	54,493.02	57,181.50
grassland (ha)	39,998.70	39,593.25	39,597.48	39,592.62
water bodies (ha)	105,536.34	106,114.32	106,021.08	106,030.17
construction land (ha)	204,083.01	227,297.70	231,223.59	225,642.60
unused land (ha)	24.48	24.39	24.39	23.94
carbon emissions (ten thousand tons)	3493.11	7578.11	7728.05	7518.24
carbon accumulation (ten thousand tons)	7507.06	7694.08	7824.75	7920.27
ecological benefits (CNY one hundred million)	249.18	249.29	250.23	251.24
economic benefits (CNY one hundred million)	8101.87	18,278.64	32,166.54	54,506.56

## Data Availability

The data presented in this study are available upon reasonable request from the corresponding author. The data are not publicly available due to consent provided by participants.

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
