# Peer review of "Optimization of the Territorial Spatial Patterns Based on MOP and PLUS Models: A Case Study from Hefei City, China"

_ijerph, 2023, doi:10.3390/ijerph20031804_

Round 1
Reviewer 1 Report
This paper presents interesting insights about spatial planning in a specific context of Chinese city. I am convinced that such research is very usable both from the perspective of scientific sector and (even more) business/planning sector. Moreover, the case city seems to be a good example to study. Structure of this manuscript is appropriate, but I have some questions and suggestions, I will report on these in detail below.
l.126: “Socio-economic and energy data”, What statistics were used are not specified in the article.
l.201: “3.1.1. Optimization schemes” reads still like methods; similar comment concerns to the section 3.1.2. (starting in l. 243). Please clarify this and correct if necessary.
l.203: Why choose the years 2025, 2030, 2035. Are these in line with the 5 year plans?
l.237: Table 1 seems to have many values repeated. Is this logical? Except construction land, why carbon emission coefficient, carbon accumulation coefficient and ecological benefit coefficient do not differ in years 2025, 2030 and 2035?
l.296: Table 2. After 2030, the growth of cultivated land and forest land is obvious, and the decrease of construction land is obvious, even lower than the value in 2025. Is it real?
l.323-325: How does the author understand the “proximity factor”? And why use these “distance” to characterize proximity factors? It should be explained clearly.
l.339: Similar to the previous suggestion, why does the author select these specific POI in the study?
Reviewer 2 Report
The article addresses the problem of the optimization of territorial spatial scale in the study case of Hefei City, China. The proposed approach implements a multi-objective optimization programming (MOP) model and visualizes data results using Geographical Information System (GIS) software. The optimization functions are based on Minimum carbon emission, Maximum carbon accumulation, Maximum ecological benefits, and Maximum economic benefits. Data instances were used from remote sensing images of Hefei City from 2010 and 2018, acquired from the Resource and Environment Science and Data Center of the Chinese Academy of Sciences (http://www.resdc.cn). The development platform to implement the MOP method was LINGO12, and the data visualization was in ArcGIS 10.2 software. Experimental results showed that the area of forest land, a significant ecological land, would continue to grow with a large increment. However, the area of unused land and those covered by grassland and water bodies would reach a stable state with some fluctuations with minor changes. Therefore, the authors conclude that after 2030, the construction land area would change from decreased to negative growth, while the net area of farmland would start to increase, and carbon emissions would peak in 2030.
1. Line 11. It is recommended to simplify the first sentence; it is hard to understand the idea behind a sentence in more than three lines. It is the same case in line 91.
2. It is recommended to provide quantitative results from the multi-objective model compared with other approaches in the abstract and section one.
3. Describing the article organization [sections] in section one is recommended.
4. Line 94. It is recommended to define the acronymous before it is used, e.g., GIS, CLUE-S, CA, etc.
5. Results do not present a comparative table of Pareto dominance front to verify the quality of solutions. Also, there needs to be a comparison of solutions diversity by the MOP approach to compare diversity. Therefore, it is impossible to conclude which solution is optimal for the multi-objective model presented. Metrics used to evaluate the multi-objective algorithm that authors could present are error ratio, spacing, spread, hypervolume and relative hypervolume, attainment surface, etc. It is crucial to compare metrics used to evaluate multi-objective algorithms to determine the quality of solutions and computational efficiency. Computational performance [simulations] are not complete for the multi-objective model presented. In this way, it is hard to corroborate the quality of solutions in a quantitative way of the proposed MOP approach.
6. It is recommended to present a pseudocode or a flowchart about the multi-objective method approach used for replication of other authors. Also, it is essential to present an experimental analysis in deep of the proposed MOP approach.
Authors should pay attention to long sentences in the document.
Reviewer 3 Report
This paper presented a methodology that focuses on ecology and economic optimization to meet the goal of minimal carbon emission, maximum carbon accumulation, ecological benefits, and economic benefits.
The manuscript is appealing because a spatial modeling approach is used to achieve multi-objective prediction, and optimization of the territorial spatial layout. Under the “double carbon” target of peak and neutrality, the simulation results could be a reference for cities in China and other developing countries.
However, several points are missing, or not well presented, rendering a minor revision suggestion before this manuscript can be accepted for publication.
The “Minimum carbon emission “ of the methodology was not well described. I understand that the optimization schemes of section 3.1.1 might be from other literature, but as this is the basis of the study, I strongly encourage the authors to describe it here.
Figure 4: it’s extremely hard to tell the difference between the three scenarios, please either optimize the figure or the results.
The methodology section does not cover the weights of spatial driving factors, each individual factor will have a positive or negative effect. How do you justify their impact? Do you consider their weight?
Specific comments:
Line 206 to 236 please try to avoid such bullet points in scientific writing
Line 239-292 This section includes several equations and definitions which make it hard to understand. I suggest the authors optimize the display of this section.
Round 2
Reviewer 2 Report
A multi-objective method aims to approximate the Pareto front [best quality solutions] to cover the Pareto front [devirsity, compromises, etc.]. Metrics must evaluate the quality of results [Pareto front nearest solutions], diversity of finding solutions, the number of elements of the Pareto set, and computational efficiency of the method used [algorithm]. It is impossible to compare the obtained results directly, for it only sometimes is possible to use all metrics, but it is necessary to use at least one metric to justify the results obtained. This way, metrics are used to justify the best solutions for the problem addressed. If Lingo software does not permit comparing solutions in a Pareto front, it is recommended to save in a File the solutions obtained for any objective function of the problem, and then you could use Excel software to create the Pareto front charts. Here is a link for the Pareto front chart using Excel: https://support.microsoft.com/en-us/office/create-a-pareto-chart-a1512496-6dba-4743-9ab1-df5012972856
Pareto Front is a set of nondominated solutions chosen as optimal if no objective can be improved without renouncing at least one other objective. The Pareto front gives the solution for multi-objective optimization. Furthermore, the Pareto chart gives a set of optimal solutions that are non-dominating in nature. Thus, a Pareto chart must ensure the best solutions obtained by LINGO software to justify your experimental results in the article.
Author Response
Thank you very much for your suggestions again. According to your suggestions and the links provided, we have created the Pareto charts of the solution of each function.